# Clinical utility and acceptability of a whole-hospital, pro-active electronic paediatric early warning system (the DETECT study): A prospective e-survey of parents and health professionals

**Bernie Carter**[1]*, **Holly Saron**[1], **Lucy Blake**[2], **Chin-Kien Eyton-Chong**[3], **Sarah Dee**[4], **Leah Evans**[4], **Jane Harris**[5], **Hannah Hughes**[6], **Dawn Jones**[7], **Caroline Lambert**[8,9], **Steven Lane**[10], **Fulya Mehta**[3], **Matthew Peak**[11], **Jennifer Preston**[12], **Sarah Siner**[7], **Gerri Sefton**[13‡], **Enitan D. Carrol**[8,9‡]

1 Faculty of Health, Social Care and Medicine, Edge Hill University, Ormskirk, United Kingdom, 2 Department of Social Sciences, University of West of England, Bristol, United Kingdom, 3 Department of General Paediatrics, Alder Hey Children's NHS Foundation Trust, Liverpool, United Kingdom, 4 High Dependency Unit, Alder Hey Children's NHS Foundation Trust, Liverpool, United Kingdom, 5 Faculty of Health, Public Health Institute, Liverpool John Moores University, United Kingdom, 6 Oncology Unit, Alder Hey Children's NHS Foundation Trust, Liverpool, United Kingdom, 7 Clinical Research Division, Alder Hey Children's NHS Foundation Trust, Liverpool, United Kingdom, 8 Institute of Infection, Veterinary and Ecological Sciences, University of Liverpool, Liverpool, United Kingdom, 9 Department of Infectious Diseases, Alder Hey Children's NHS Foundation Trust, Liverpool, United Kingdom, 10 Institute of Translational Medicine, University of Liverpool, Liverpool, United Kingdom, 11 NIHR Alder Hey Clinical Research Facility, Alder Hey Children's NHS Foundation Trust, Liverpool, United Kingdom, 12 Department of Women's and Children's Health, Institute of Life Course and Medical Sciences, University of Liverpool, Liverpool, United Kingdom, 13 Paediatric Intensive Care Unit, Alder Hey Children's NHS Foundation Trust, Liverpool, United Kingdom

‡ GS and EDC were the Co-Chief Investigators.
* bernie.carter@edgehill.ac.uk

**Data Availability Statement:** The North-West, Liverpool East Research Ethics Committee placed

## Abstract

### Background

Paediatric early warning systems (PEWS) are a means of tracking physiological state and alerting healthcare professionals about signs of deterioration, triggering a clinical review and/or escalation of care of children. A proactive end-to-end deterioration solution (the DETECT surveillance system) with an embedded e-PEWS that included sepsis screening was introduced across a tertiary children's hospital. One component of the implementation programme was a sub-study to determine an understanding of the DETECT e-PEWS in terms of its clinical utility and its acceptability.

### Aim

This study aimed to examine how parents and health professionals view and engage with the DETECT e-PEWS apps, with a particular focus on its clinical utility and its acceptability.

ethical restrictions on sharing the data publicly. The anonymised datasets used and/or analysed during the survey are available from the corresponding author on reasonable request. Additionally the data underlying the results presented in the study are available from Prof Chris Littlewood (external to research team) Chris.Littlewood@edgehill.ac.uk.

**Funding:** NIHR Invention for Innovation i4i Programme. https://fundingawards.nihr.ac.uk/award/II-LA-0216-20002 Initials of authors:EDC, GS, BC, MP. The funders had no role in study design, data collection and analysis, decision to publish, or preparation of the manuscript.

**Competing interests:** The authors have declared that no competing interests exist.

## Method

A prospective, closed (tick box or sliding scale) and open (text based) question, e-survey of parents (n = 137) and health professionals (n = 151) with experience of DETECT e-PEWS. Data were collected between February 2020 and February 2021.

## Results

Quantitative data were analysed using descriptive and inferential statistics and qualitative data with generic thematic analysis. Overall, both clinical utility and acceptability (across seven constructs) were high across both stakeholder groups although some challenges to utility (e.g., sensitivity of triggers within specific patient populations) and acceptability (e.g., burden related to having to carry extra technology) were identified.

## Conclusion

Despite the multifaceted nature of the intervention and the complexity of implementation across a hospital, the system demonstrated clinical utility and acceptability across two key groups of stakeholders: parents and health professionals.

## Introduction

Paediatric early warning systems (PEWS) encompass a range of different interventions [1]. They are a means of tracking physiological state and alerting healthcare professionals about signs of deterioration, triggering a clinical review and/or escalation of care of children [2]. PEWS are reported to be used extensively internationally [2, 3] and across different health care settings such as emergency departments [4–6], oncology and haematology [7–9], and more rarely, hospital wide [10, 11] or nationally [12]. PEWS are used in paediatric in-patient hospital settings [2] in resource-rich [10] and resource-limited countries [7, 13]. Although electronic-based PEWS are reported as bringing additional safety benefits such as reduction in human error, greater time efficiency and instant visibility of recorded data to the clinical team [14]; this has not been reported across a whole hospital setting.

The acronym PEWS is sometimes used ambiguously in the literature to describe early warning scores [15–18] or systems [6, 10, 11, 19], or both score and system [4]. Within this paper, PEWS is used to denote system. Although PEW scores are an important step, implementing a score in isolation without considering the wider system factors [20] and socio-technical systems [2] is unlikely to be effective as it does not take into account the environment, organisational culture, policy and human action contexts which impact upon the occurrence and prevention of deterioration [10]. Smith [21] proposes a 'chain of prevention', composed of five interlinked rings of equal importance: education, monitoring, recognition, escalation, and response, as a structure for preventing and detecting patient deterioration and cardiac arrest.

Within the UK, the inquiry 'Why Children Die' report [22] led to the recommendation for "a standardised and rational monitoring system with imbedded early identification systems for children developing critical illness–an early warning score"(p4). This recommendation was made despite the evidence base for the effectiveness of PEWS being weak in terms of decreasing all-cause mortality [23] and being sufficiently sensitive in identifying children who need escalation of care in a hospital with higher levels of paediatric resource [24]. Across the UK,

use of PEW scores and systems is widespread, but a variety of scoring systems, age bandings and formats (paper and electronic) exist [25]. A recent survey identified that while there are many common elements, standardisation across the UK has yet to be achieved [1]; this standardisation is the aim of the national PEWS Programme Board [1, 25].

Within hospital settings, implementation of PEWS is complex, requiring iterative processes to sustain use [10]. This complexity, as well as the methodological challenges associated with researching effectiveness, may contribute to the weak and often conflicting evidence about whether the implementation of PEWS does lead to reductions in cardiac arrest, morbidity, and mortality [2, 10]. Effective implementation requires consideration of implementation fidelity, effectiveness, and utility and account needs to be taken of key components of the system such as situational awareness [6, 20, 26], communication [7], the interface of the system with the users [27], the degree of change to workflow [19], the barriers and enablers of uptake [28, 29], and embedding and adaptation over time [10].

This paper reports the findings from survey data generated as part of one of the sub-studies from the Dynamic Electronic Tracking and Escalation to reduce Critical Care Transfers (DETECT) study [30].

## The DETECT surveillance system

The DETECT study implemented a proactive end-to-end deterioration solution (the DETECT surveillance system, Fig 1) across a tertiary children's hospital. This built on earlier work on translating PEW scoring from paper to electronic surveillance [14]. The DETECT surveillance system aims to proactively screen paediatric patients for early signs of serious deterioration or sepsis, thereby reducing complications and emergency transfers to critical care following deterioration in hospital.

The DETECT surveillance system is supported by System C's, CareFlow Connect and Vitals (paediatric version) apps. Vitals is an electronic observation and decision support system, which involves staff using an electronic hand-held device (iPod touch in this study) to record children's vital signs. The recorded signs include breathing rate, effort of breathing, oxygen saturation, oxygen requirement, heart rate, blood pressure, capillary refill time, temperature, 'alertness, verbal responsiveness, pain responsiveness, or unresponsiveness (APVU)', and nurse or parental concerns (Fig 2). The recorded data automatically calculate a pre-defined PEW score, which categorises the risk (low, moderate, critical) of developing serious illness. CareFlow Connect is an encrypted communication system, which interacts with Vitals to provide automated alerts about the sickest children, generated from the PEW score or suspicion of sepsis, and includes the ability to escalate concerns direct to the clinical team who can respond in real-time, without the nurse leaving the child's bedside. These modified apps are referred to as DETECT e-PEWS and are used by health professionals using iPods to document vital signs or respond to alerts of deterioration triggered by the system.

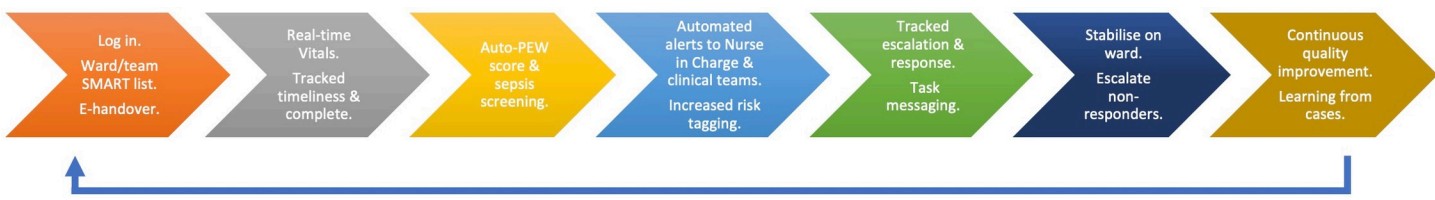

**Fig 1. DETECT surveillance system.**

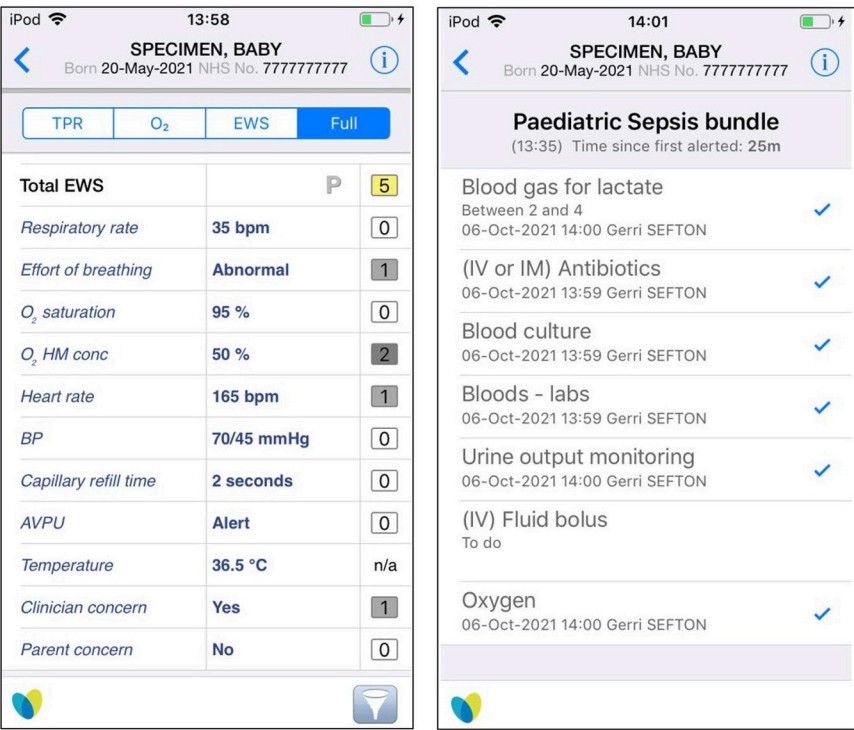

**Fig 2. Example screenshots from iPod touch: DETECT e-PEW score screen and sepsis bundle overview (fictitious patient data).**

Both apps had bespoke modifications made to them for the purpose of the DETECT study. The PEW score thresholds used the established Alder Hey age-specific PEW score and proactive screening for early signs of sepsis used modified NICE criteria [31].

Hands-on training in using the DETECT system, and education about the rationale for introducing the system (e.g., reducing human error in calculating scores, and reducing deterioration and need for transfer to critical care) was delivered to all health professionals who would be using the system, either in small groups or one-to-one. Children and parents were made aware of the implementation of the technology using dedicated posters in all public areas and an explanation provided at the child's admission. Following staff training the apps were deployed on iPod touch and iPads across ten in-patient wards (240 beds). Each member of ward staff providing direct clinical care to children carried an iPod, and the Nurse in Charge of the shift had an iPad for overview of the entire ward. The clinical teams had a minimum of one iPod per team; each member of the on-call team (medical team that provides out-of-hours cross cover for inpatients and new admissions) and those within the Acute Care Team (nurse-led Rapid Response Team) each had an iPod or iPad (some used their personal mobile phone). Additionally, there was an agreement, approved by Trust Information Governance, that staff could have the Careflow Vitals and Connect apps loaded to their personal mobile phone under the 'Bring Your Own Device' scheme which meant that some staff did not have to carry an additional device and it was more convenient for them (Vitals is device specific (Apple), Connect is device agnostic and works on all personal devices).

The vital signs data were visible in real-time on iPods, iPads, computers and personal devices and were also integrated back to Meditech, the electronic patient record (EPR) used by the study hospital.

In the study setting, Vitals was implemented as a mandatory practice for monitoring all in-patients in study wards, and CareFlow Connect was implemented a month later, and was available but not mandatory, and its use was inconsistent.

### Defining clinical utility and acceptability

It is important to define the concepts of clinical utility and acceptability, as it is evident in the literature that there is ambiguity and overlap in what is encompassed by the terms, and there is no consensus on definitions [32]. Within this discussion, clinical utility is defined in its narrowest sense; does the technology do what it is supposed to do, and does it perform its designated function [33]?

However, the complexity inherent in implementation, adoption and assimilation of technology in healthcare systems [34, 35] requires the definition of acceptability to acknowledge its multifaceted nature, and to be more encompassing [32, 34, 36, 37]. The Theoretical Framework of Acceptability (TFA) (v2) [37] is composed of seven component constructs (Fig 3): 'affective attitude', 'burden', 'ethicality', 'intervention coherence', 'opportunity costs', 'perceived effectiveness' and 'self-efficacy'. The TFA proposes that acceptability is a *"multi-faceted construct that reflects the extent to which people delivering or receiving a healthcare intervention consider it to be appropriate, based on anticipated or experienced cognitive and emotional responses to the intervention"* [37].

The aim of this part of the sub-study was to generate a broad, baseline understanding of the DETECT system in terms of its clinical utility and its acceptability to health professionals who had experience of using the handheld (iPods and iPads) DETECT e-PEWS and to parents and children who had received care by professionals using the system. The research question underpinning this sub-study was: 'How do parents and health professionals view and engage with the DETECT e-PEWS?'

The Consensus-Based Checklist for Reporting Survey Studies (CROSS) [38] has been used to ensure high quality reporting.

## Materials and methods

### Study design

A prospective e-survey (paper copies available if preferred) using closed (tick box or sliding scale) and open (text based) questions.

### Participants and setting

The target population was parents of children (aged 0–18 years old) who were in-patients (excluding children admitted as day-cases, the paediatric intensive care unit or neonatal surgical unit), and health professionals at Alder Hey Children's Hospital, a tertiary setting in Liverpool in the UK. No sample size calculation was used.

| **Construct 1**<br>**Affective attitude**<br>Most parents & health professionals felt positively about the system. | **Construct 2**<br>**Burden**<br>Most parents & health professionals did not perceive the system to be burdensome | **Construct 3**<br>**Ethicality**<br>Most parents & health professionals perceived the system to fit with their values. | **Construct 4**<br>**Intervention coherence**<br>Most parents & health professionals understood the system and how it worked. | **Construct 5**<br>**Opportunity costs**<br>Most parents & health professionals accepted the required changes resulting from the system. | **Construct 6**<br>**Perceived effectiveness**<br>Most parents & health professionals perceived the system was likely to achieve its purpose. | **Construct 7**<br>**Self-efficacy**<br>Most parents & health professionals felt confident about engaging with the system. |

**Fig 3. Domains of theoretical framework of acceptability (v2) as applied to findings.**

Recruitment of parents was undertaken face-to-face either by a researcher or, during the COVID-19 pandemic, by dedicated, trained DETECT study research nurses. Recruitment occurred between April 2020 and February 2021, although recruitment (staff shortages) was not possible in some months). A mixture of convenience, purposive and snowball sampling was used. Two groups of parents were recruited: Group 1 (parents whose children had not experienced a critical deterioration event during admission; non-CDE group) (n = 68 parents), and Group 2 (parents whose children had experienced a critical deterioration event, CDE group) (n = 69 parents). A CDE was defined as a deterioration where the patient is critically unwell, which culminates in an emergency transfer to high dependency unit or the intensive care unit, or an unexpected death.

Recruitment of health professionals (doctors, nurses and allied health professionals, n = 151) with experience of using the system was initially opportunistically face-to-face on the wards by the trained research nurses asking staff if they were interested and latterly by email. Recruitment occurred between May 2020 and January 2021; face-to-face recruitment only occurred across seven months due to the pandemic (staff shortages). Health professionals were also given detailed information via information sheets, given sufficient time to consider if they wanted to participate and the opportunity to ask questions (face-to-face or remotely). The possibility of coercion was avoided by making it clear that participation was voluntary and leaving the device with the survey link on it with the potential participant for about ten minutes; allowing the participant to complete the survey or not, as preferred.

Parents were approached on the wards where the DETECT devices (iPods or iPads) were being used and asked if they were interested in the study. Tailored information sheets for parents were given to potential participants. Consent by parents and health professionals for participation in the survey was gained via a 'tick box' at the start of the survey.

## Parent involvement and engagement

To ground the design and content of the survey and to ensure that the wording and flow of the questions were clear and unambiguous parents were engaged with via two face-to-face workshop groups (n = 8) and by email (n = 3) from the Alder Hey Children's NHSFT Parent and Carer's Research Forum which is funded by the National Institute for Health Research (NIHR) Alder Hey Clinical Research Facility (CRF). Potentially sensitive questions such as whether parents were able to identify if their child was deteriorating were discussed with the parents and the final wording used and its positioning at the end of the survey were both informed by discussion with the parents. These contributions and the refinements were made to the survey were helpful in creating an engaging and sensitive survey that was well-received by parents.

## E-surveys

Semi-structured surveys (non-validated) were specifically designed for the study, these survey instruments were not validated and, as previously noted, their sensitivity in terms of health literacy had been checked with parents. Consultation with health professionals (paediatric doctors and nurses who were part of our wider steering group) helped to develop the structure, content, and readability of the health professional survey. Pretesting/piloting of our proposed final versions of the surveys was carried out with parents (see engagement in previous section, n = 11) and health professionals (nurses and doctors, n = 5) was carried out on one occasion; no revisions were identified as being required. Closed (core) questions required mandatory responses to avoid non-response error. None of the questions were weighted. Three versions, each taking about 3–5 minutes to complete, were created one for health professionals, one for parents of children who had experienced a critical deterioration event (CDE), one for non-

CDE parents. In the parent surveys we referred to vital signs as 'obs' (an abbreviation of observations).

**Parent survey.** Questions in the non-CDE and CDE parent surveys were identical apart from one additional question for CDE parents. The surveys were only available in English and no dedicated translation was available. The surveys consisted mainly of closed (tick box or sliding scale) questions (n = 13) some with more than one item; three questions had a box for parents to provide further comments. There was one open question at the end of both the CDE and non-CDE parent surveys. The surveys designed for parents were composed of 5 sections: (1) Introduction (brief information about the survey and the study); (2) Deciding to Take Part (consent); (3) Background Information (n = 5 questions asking about relationship to child, gender and age of their child, ward their child is/was being treated on, number of times their child has been admitted to the study hospital); (4) About the Device (n = 9 questions asking about satisfaction with explanations about the device and how it was used, trust in technology, feeling safe and secure with the device (relating to our definitions of utility [33] and acceptability [37]). A final question asked parents if they know when their child is getting 'poorlier' (deteriorating); and a (5) Thank you section. Typically, the researcher did not assist parents to complete the survey, although support was available, as needed.

**Health professional survey.** This consisted of closed (n = 21) questions (drop down response, tick box or sliding scale) some with more than one item; all but five questions had a comment box. The health professionals' survey consisted of eight sections, five of which (education, monitoring, recognition, escalation, and response) related to the chain of prevention [21]: Introduction; Deciding to Take Part; Education and Training, (n = 2 questions); Monitoring and Recognition (n = 5 questions); Escalation and Response (n = 3 questions); System Features (n = 4 questions), Concluding Thoughts (n = 3 questions); all but the first section related to our definitions of utility [33] and acceptability [37].

## Ethics

The study gained ethics approval via the North-West, Liverpool East Research Ethics Committee (IRAS ID: 215339). All those involved in gaining consent were suitably qualified, experienced, and trained and consent was gained in accordance with the principles of Good Clinical Practice on Taking Consent [39]. No potential participant was put under any level of pressure and their right to refuse to participate in the survey without giving reasons was respected. All relevant governance protocols relating to data management and anonymisation were followed.

Participants ticked a consent/assent box at the start of the survey and submission of the surveys was taken as confirming consent (or assent) to participate in the study. All responses were anonymous unless they chose to share their contact details for potential participation in next phase (interview) of the study. Direct feedback to individual survey participants was not possible (due to anonymity of survey responses) but findings will be shared with the broad population of parents through the Parent and Carer's Research Forum, hospital newsletter, social media etc. and with health professionals via Grand Rounds and other meetings. All required data governance procedures were followed.

## Analysis

The survey responses were coded and analysed using descriptive and inferential statistics within a statistical package (SPSS v25). The text in the open questions was collated and subjected to generic, descriptive thematic analysis. The results are reported separately for the children, parents (non-CDE and CDE), and health professionals (depending on role).

Descriptive statistics, mean (M) and standard deviation (SD), are presented to describe variables measured on a continuous scale, categorical variables are reported using counts and percentages. For the health professional data, Chi-squared and Fishers exact test were used to assess between group differences when the outcome of interest was categorical and independent T-test was used when outcome was continuous.

## Results

### Characteristics of participants

**Parents.** Of the parents approached, there was a 9–10% decline rate (typical reasons for declining being focused on child). In total, 137 parents completed the survey (mothers n = 115, 83.9% and fathers n = 22, 16.1%); of these, there were 68 non-CDE parents (n = 59 mothers, n = 9 fathers) and 69 CDE parents (n = 56 mothers, n = 13 fathers) (Table 1). Around half the parents (n = 27 CDE and n = 38 non-CDE) provided open text responses. All but three surveys (n = 134) were completed electronically.

The parents reported on the experiences of their sons (n = 79, 57.7%) and daughters (n = 58, 42.3%). The age range was < 1 year-13 years or older with the majority (n = 67, 48.9%) being in the < 1 year category. For most of the children (n = 83, 60.6%) this was their first admission, although 11 children (8%) had experienced ten or more admissions. There was representation across all eligible ward settings with most children across both groups nursed on the cardiac unit (n = 29) and general paediatrics (n = 27) at the time of survey completion. Focusing solely on CDE children, most were nursed on the cardiac unit (n = 22), high dependency unit (n = 13), and general paediatrics (n = 10) at the time of survey completion (Table 1).

**Health professionals.** In total 151 health professionals participated in the survey (decline rate not calculated as staff were approached by email as well as directly but typical reason for declining being 'too busy'). Of the 151 participants, the majority (n = 102, 67.5%) had been using DETECT e-PEWS for 6 months or longer, with 49 (32.5%) having used the device (iPod or iPad) for <6 months. Forty four percent (n = 66) of HPs provided at least one open text, with just under half of these (n = 25) at least three open text responses; some provided up to nine. All surveys were completed electronically.

The sample included nurses, doctors and allied health professionals who were using DETECT e-PEWS in two distinct ways and the data are grouped and presented in this way: 'Documenting Vital Signs (D-VS) which involved the work of taking and recording the child's vital signs into the app on the iPod or 'Responding to Vital Signs' (R-VS) which encompassed the work of responding to concerns and alerts on the iPods, iPads or personal device from the automatically generated PEWS scores and taking appropriate action (Table 2).

In the D-VS group (n = 133) the disciplinary role of the participants was Staff Nurse (n = 78, 51.7%), followed by Sister (n = 19, 12.6%), Student Nurse (n = 16, 10.6%), Allied Health Professional (n = 10, 6.6%), Assistant Practitioner (n = 2, 1.4%) Ward Manager (n = 1, 0.7%).

In the R-VS group (n = 18) the reported role of most participants was Doctor (n = 14, 9.3%), Advanced Clinical Practitioner (n = 2, 1.3%), and Acute Care Team (n = 2, 1.4%).

Health professionals worked across all 10 of the eligible ward settings with the majority working on four wards: general paediatrics (n = 34, 22.5%), medical speciality (n = 29, 19.2%), neurology (n = 26, 17.2%) and the high dependency unit (n = 21, 13.9%) (Table 3).

### Parents: Core findings

Data have been reported from parents in two groups: parents whose children had not experienced a critical deterioration event (non-CDE) and those whose children who had (CDE). Labels are used to indicate parent number from survey and whether parent was CDE or

**Table 1. Parent and child demographics from parent survey responses.**

| | non-CDE | | CDE | |
|---|---|---|---|---|
| **Parent status** | N (68) | % | N (69) | % |
| Mother | 59 | 86.8 | 56 | 81.2 |
| Father | 9 | 13.2 | 13 | 18.8 |
| **Child Gender** | | | | |
| Girl | 29 | 42.6 | 29 | 42 |
| Boy | 39 | 57.4 | 40 | 58 |
| **Child Age** | | | | |
| < 1 year | 23 | 33.8 | 44 | 63.8 |
| 1 - < 2 years | 5 | 7.4 | 6 | 8.7 |
| 2 - < 7 years | 19 | 27.9 | 9 | 13 |
| 7 - < 13 years | 12 | 17.6 | 7 | 10.1 |
| >13 years | 9 | 13.2 | 3 | 4.3 |
| **Number of Admissions** | | | | |
| First admission | 43 | 63.2 | 40 | 58 |
| 2–5 admissions | 14 | 20.6 | 17 | 24.6 |
| 6–10 admissions | 4 | 5.9 | 8 | 11.6 |
| >10 admissions | 7 | 10.3 | 4 | 5.8 |
| **Ward** | | | | |
| Cardiac | 7 | 10.3 | 22 | 31.9 |
| General paediatrics | 17 | 25 | 10 | 14.5 |
| General surgery | 10 | 14.7 | 5 | 7.2 |
| High dependency unit* | 1 | 1.5 | 13 | 18.8 |
| Medical speciality | 11 | 16.2 | 3 | 4.3 |
| Oncology | 6 | 8.8 | 7 | 10.1 |
| Speciality surgery | 10 | 14.7 | 3 | 4.3 |
| Neurology | 5 | 7.4 | 5 | 7.2 |
| Burns | 1 | 1.5 | 0 | 0 |
| Emergency decision unit** | 0 | 0 | 0 | 0 |

* The high dependency unit (HDU) provides level 2 critical care [40]. The HDU patient population includes patients who have deteriorated on the ward, high acuity patients post-operatively as well as some step-downs from PICU with higher care needs than can be delivered on a ward.

** EDU is a short stay unit of admissions direct from ED who either stabilise and are discharged or are admitted to another ward within 24 hours.

non-CDE, for example, (CDE P12). Overall, the parents in both groups had similar experiences in terms of their engagement with and perceptions of the devices (Table 4). Most parents reported that they know when their child is 'getting poorlier' either "all' or 'some' of the time; non-CDE (n = 63, 92.6%) and CDE (n = 62, 89.8%). Summary statistics are reported in Table 4.

**Overall satisfaction.** On a scale of 0 to 100, most parents indicated high levels of satisfaction with the devices (M scores: non-CDE parents 86%, CDE 89%).

**Initial impressions.** Most parents (non-CDE 93%, CDE 81%) noticed the nurses using a device to do their child's vital signs. Over half of the parents recalled that the person taking and recording their child's vital signs had explained the device to them (CDE 52%, non-CDE 59%). One CDE parent explained:

**Table 2. Aspect of DETECT e-PEWS app used and professional role.**

| Aspect of DETECT e-PEWS app and role | N | % |
|---|---|---|
| **Documenting vital signs (D-VS) on iPod** | 133 | 88.1 |
| Staff Nurse | 78 | 51.7 |
| Sister | 19 | 12.6 |
| Student Nurse | 16 | 10.6 |
| Allied Health Professional* | 10 | 6.6 |
| Health Care Assistant | 8 | 5.3 |
| Ward Manager | 1 | 0.7 |
| Assistant Practitioner** | 2 | 1.4 |
| **Responding to vital signs (R-VS) on iPad** | 18 | 11.9 |
| Doctor | 14 | 9.3 |
| Advanced Clinical Practitioner*** | 2 | 1.3 |
| Acute Care Team**** | 2 | 1.4 |
| **Length of time using DETECT e-PEWS** | | |
| <6 months | 49 | 32.5 |
| ≥ 6 months or longer | 102 | 67.5 |

* Allied Health Professional is a term that includes physiotherapists and occupational therapists. We did not collect data on the specific profession of AHPs.

**Assistant Practitioners are not registered practitioners but they support care and have a high level of skill through their experience and training [41].

*** Advanced Clinical Practitioners are nurses or AHPs trained to Masters level on an approved ACP course who deliver clinical caseload management autonomously to acute and complex patient groups [42].

**** Acute Care Team is the nurse led Rapid Response Team in the study hospital.

> *Initially, I had no clue what the device was for and did wonder if nurses were on their phones but now I know what they were doing I have been happy for them to use it (CDE P12).*

Initially, about a third (non-CDE 40%, CDE 33%) thought the device was the professional's own phone. Similarly, around 40% of parents (non-CDE 40%, CDE 44%), were initially unsure about the purpose of the device; although at the time of filling in the survey, most understood the purpose of the device (non-CDE 71%, CDE 65%).    **Technology related.**    Most parents (non-CDE 85%, CDE 88%) agreed that 'improvements in technology are a good thing'. One parent noted that the *"device seems to make 'obs' quicker"* (CDE, P46) with another noting it was *"wonderful for speed and efficiency. . .and a great observation checklist for the nurses"* (CDE

**Table 3. Ward/unit professionals working on.**

| Ward | N | % |
|---|---|---|
| General paediatrics | 34 | 22.5 |
| Medical speciality | 29 | 19.2 |
| Neurology | 26 | 17.2 |
| High dependency unit | 21 | 13.9 |
| General surgery | 11 | 7.3 |
| Specialist surgery | 11 | 7.3 |
| Oncology | 8 | 5.3 |
| Cardiac | 6 | 4.0 |
| Burns | 3 | 2.0 |
| Emergency decision unit | 2 | 1.3 |

**Table 4. Parents' responses to survey.**

| | non-CDE Parent | | CDE Parent | |
|---|---|---|---|---|
| | **N (68)** | **%** | **N (69)** | **%** |
| **Initial Impressions** | | | | |
| **When the person did your child's 'obs' did you notice them using the device?** | | | | |
| Yes | 65 | 92.6 | 56 | 81.2 |
| No | 1 | 1.5 | 3 | 4.3 |
| Can't remember | 1 | 1.5 | 7 | 10.1 |
| **To begin with I thought the person doing my child's 'obs' was on their phone** | | | | |
| Yes | 27 | 39.7 | 23 | 33.3 |
| No | 36 | 52.9 | 38 | 55.1 |
| Can't remember | 4 | 5.9 | 6 | 8.7 |
| **To begin with I didn't know what the device was doing when they were using the device** | | | | |
| Completely agree | 10 | 14.7 | 11 | 15.9 |
| Agree a bit | 17 | 25.0 | 19 | 27.5 |
| Neutral | 8 | 11.8 | 10 | 14.5 |
| Disagree a bit | 15 | 22.1 | 8 | 11.6 |
| Completely disagree | 17 | 25.0 | 19 | 27.5 |
| **I don't really understand what the device is doing** | | | | |
| Completely agree | 7 | 10.3 | 7 | 10.1 |
| Agree a bit | 6 | 8.8 | 8 | 11.6 |
| Neutral | 6 | 8.8 | 7 | 10.1 |
| Disagree a bit | 9 | 13.2 | 8 | 11.6 |
| Completely disagree | 39 | 57.4 | 37 | 53.6 |
| **Did the person doing your child's 'obs' explain what the device was for?** | | | | |
| Yes | 35 | 51.5 | 41 | 59.4 |
| No | 28 | 41.2 | 19 | 27.5 |
| Can't remember | 4 | 5.9 | 7 | 10.1 |
| **Technology Related** | | | | |
| **Improvements in the technology are a good thing** | | | | |
| Completely agree | 58 | 85.3 | 61 | 88.4 |
| Agree a bit | 5 | 7.4 | 5 | 7.2 |
| Neutral | 1 | 1.5 | 1 | 1.4 |
| Disagree a bit | 1 | 1.5 | - | - |
| Completely disagree | 1 | 1.5 | - | - |
| **I don't trust technology like this** | | | | |
| Completely agree | 2 | 2.9 | | |
| Agree a bit | 3 | 4.4 | 3 | 4.3 |
| Neutral | 8 | 11.8 | 5 | 7.2 |
| Disagree a bit | 12 | 17.6 | 11 | 15.9 |
| Completely disagree | 42 | 61.8 | 48 | 69.6 |
| **The person using the device sometimes has problems with it** | | | | |
| Completely agree | 3 | 4.4 | 1 | 1.4 |
| Agree a bit | 11 | 16.2 | 8 | 11.6 |
| Neutral | 22 | 32.4 | 21 | 30.4 |
| Disagree a bit | 6 | 8.8 | 12 | 17.4 |
| Completely disagree | 25 | 36.8 | 24 | 34.8 |
| **There are always enough devices available when the person needs to do my child's 'obs'** | | | | |
| Completely agree | 35 | 51.5 | 32 | 46.4 |

(*Continued*)

**Table 4.** (Continued)

| | non-CDE Parent | | CDE Parent | |
|---|---|---|---|---|
| | **N (68)** | **%** | **N (69)** | **%** |
| Agree a bit | 9 | 13.2 | 9 | 13.0 |
| Neutral | 20 | 29.4 | 23 | 33.3 |
| Disagree a bit | 1 | 1.5 | 2 | 2.9 |
| Completely disagree | 2 | 2.9 | 1 | 1.4 |
| **Engagement with health professional** | | | | |
| **My child doesn't mind the person using the device to record their 'obs'** | | | | |
| Completely agree | 55 | 80.9 | 55 | 79.7 |
| Agree a bit | 5 | 7.4 | 1 | 1.4 |
| Neutral | 5 | 7.4 | 11 | 15.9 |
| Disagree a bit | - | - | - | - |
| Completely disagree | 1 | 1.5 | - | - |
| **The person doing my child's 'obs' just concentrates on the device then goes away** | | | | |
| Completely agree | 7 | 10.3 | 3 | 4.3 |
| Agree a bit | 8 | 11.8 | 12 | 17.4 |
| Neutral | 11 | 16.2 | 12 | 17.4 |
| Disagree a bit | 12 | 17.6 | 9 | 13.0 |
| Completely disagree | 29 | 42.6 | 31 | 44.9 |
| **After they've been done, I 'd like to be able to see the results of my child's 'obs'** | | | | |
| Completely agree | 28 | 41.2 | 31 | 44.9 |
| Agree a bit | 10 | 14.7 | 10 | 14.5 |
| Neutral | 23 | 33.8 | 20 | 29.0 |
| Disagree a bit | 4 | 5.9 | 1 | 1.4 |
| Completely disagree | 2 | 2.9 | 5 | 7.2 |
| **Feeling safe** | | | | |
| **I like the idea that an automated alert will be sent to a senior nurse or doctor is the device detects something of concern** | | | | |
| Completely agree | 62 | 91.2 | 65 | 94.2 |
| Agree a bit | 2 | 2.9 | 2 | 2.9 |
| Neutral | 1 | 1.5 | - | - |
| Disagree a bit | 1 | 1.5 | - | - |
| Completely disagree | 1 | 1.5 | - | - |
| **I feel safe knowing that the device aims to provide backup to the doctors and nurses** | | | | |
| Completely agree | 54 | 79.4 | 61 | 88.4 |
| Agree a bit | 11 | 16.2 | 5 | 7.2 |
| Neutral | 1 | 1.5 | 1 | 1.4 |
| Disagree a bit | - | - | - | - |
| Completely disagree | 1 | 1.5 | - | - |
| **I don't trust technology like this** | | | | |
| Completely agree | 2 | 2.9 | - | - |
| Agree a bit | 3 | 4.4 | 3 | 4.3 |
| Neutral | 8 | 11.8 | 5 | 7.2 |
| Disagree a bit | 12 | 17.6 | 11 | 15.9 |
| Completely disagree | 42 | 61.8 | 48 | 69.6 |
| **Based on knowing my child I know when they are getting poorlier** | | | | |
| All of the time | 42 | 61.8 | 36 | 52.2 |
| Some of the time | 21 | 30.9 | 26 | 37.7 |
| Not very much of the time | 4 | 5.9 | 4 | 5.8 |

P57). Typical responses included parents thinking that the technology *"lowers the risk of mistakes being made when using paper"* (CDE P3), delivers the *"right results we need to know about her"* (CDE P9) and noting that if HPs *"do obs on paper they can lose paper obs and have to do them again"* (non-CDE P11). A non-CDE parent noted that they thought that:

> *. . .. the device is a good idea, anything that ensures all the necessary people are seeing his obs has got to be a good thing in my opinion! (non-CDE P2).*

Most parents agreed (non-CDE 65%, CDE 59%) that there were always enough devices available when needed and most (non-CDE 79%, CDE 85%) disagreed with the statement that 'I do not trust technology like this'.

**Engagement with health professionals.** Most parents (non-CDE 88% and CDE 81%) agreed that their 'child did not mind the device being used to record their vital signs'. Most parents (non-CDE 60%, CDE 58%) disagreed that the 'person doing their child's vital signs just concentrated on the device and then left'. However, of those who did feel that the person doing their child's vital signs concentrated on the device and then left, one CDE parent noted that:

> *I feel when obs were taken on paper the nurse was more interactive whereas with the device they seemed to concentrate on that a lot then only let you know things were okay if prompted (CDE P24).*

However, a non-CDE parent noted:

> *Staff are nothing but interested in the patient when carrying out the obs, constantly talking and making him feel comfortable. And it's a time when he smiles the most, due to their attention and care (non-CDE P22).*

Most parents (non-CDE 56%, CDE 59%) agreed that they would have liked to have seen the results of their child's vital signs. One CDE parent noted that *"it's good to have a trace of my child's obs that isn't just paper based"* (CDE P43). One non-CDE parent expressed a need for more information:

> *I would like the nurse to talk to me more about my baby's 'obs' so that I know what I need to look for on the monitor so I could know what a SAT would mean if it went to a certain number (non-CDE P64).*

**Feeling safe.** Most parents (non-CDE 94%; CDE 97%) liked the idea that the device would trigger an automated alert if it detected something of concern. A non-CDE parent noted:

> *I feel much more at ease knowing my son's obs are going straight into the system and red flags are reviewed instantly. It's much more effective in raising concerns of poorly children. Having a complex child that deteriorates quickly and being involved in paper obs and the new technology I feel much more at ease as it's escalated much quicker (non-CDE P55).*

Most parents (non CDE 96%, CDE 96%) 'felt safe knowing the device aimed to provide a backup'. Very few parents (non CDE 7%, CDE 4%) expressed distrust in 'technology like this'; one CDE parent commented that:

*Obs are a really important part of any child's recovery, safety and definitely have shown when he's needed intervention. My experience of the obs done on the ward is that they were dealt with really quickly and efficiently which then lead to transferring to HDU. Not had any bad experiences. Couldn't of done any more than they did, they kept him safe up to the point of transfer (CDE P13).*

One parent whose child had experienced a CDE, suggested parental concern should be included as an extra safety measure (although this was already part of the system):

*My little girl's obs were not changing prior to becoming unwell so feel parental concern should also be recorded and included in 'obs' (CDE P24).*

## Health professionals: Core findings

Data have been reported from health professionals in two groups: those who documented vital signs (D-VS) using iPods and those responding to vital signs (R-VS) using iPods, iPads or personal device. Comparisons were made between groups on the continuous data using t tests. The means, standard deviations and significance levels (p values) are reported in Table 5 and the statistically significant t tests are reported in the text. Labels are used to indicate role, group and the HP number from survey, for example, (Staff Nurse, D-VS, 106).

First, the data are presented for overall satisfaction and then the remaining results are presented under headings linked to the key aspects of Smith's [21] chain of prevention.

**Overall satisfaction.** The health professionals were asked to rate their confidence and satisfaction in using the DETECT e-PEWS on a scale of 0–100. In both groups, levels of confidence and satisfaction were high. However, those in the D-VS group had significantly higher levels of confidence that they could recognise that a child's health is deteriorating than those in the R-VS group ($t$ (18, 93) = 2.46, $p$ = .024)

Similarly, the D-VS group had significantly higher levels of overall satisfaction with DETECT e-PEWS than those in the R-VS group ($t$ (17,20) = 2.82, $p$ = .012). The D-VS group also had significantly higher levels of satisfaction with being able to 'obtain a device' ($t$ (138) = -2.44, $p$ = .016). In the open-text responses, health professionals noted that *"more nursing station chargers"* (Staff Nurse, D-VS, 128) were needed and that sometimes *"people can forget to charge them"* (Sister, R-VS, 29).

**Education, training and implementation.** The D-VS group had higher levels of satisfaction 'for the education and training received' compared to the RVS group, although this difference was not statistically significant. In terms of training, there were few critical comments and these related to it being *"tricky to take in all the info and retain it for use sometimes"* (Staff Nurse, D-VS, 106) or using the device. One participant noted *"no one asked if I needed extra help, I have dyslexia"* (Staff Nurse, D-VS 48); most open responses were positive, such as:

*always someone there to help if extra advice needed (Allied Health Professional, D-VS, 2).*

Satisfaction with the 'implementation of Vitals [DETECT e-PEWS] in their area', was significantly higher in the D-VS group than the R-VS group ($t$ (17.91) = -3.46, $p$ = .003).

**Recording and monitoring.** Satisfaction was significantly higher amongst the D-VS group than the R-VS group in terms of accurately recording data ($t$ (140) = -2.08, $p$ = .040) and monitoring patients for deterioration, ($t$ (17.45) = -2.49, $p$ = .023). In relation to expectation that DETECT e-PEWS would 'reduce the incidence of omission of recording vital signs' once again the scores of the D-VS group were higher than the R-VS group, although this difference

**Table 5. Health professionals' responses: Comparison between D-Vs and R-VS\*.**

| | Documenting vital signs (D-VS) | Responding to vital signs (R-VS) | Group comparison |
|---|---|---|---|
| | M (SD) | M (SD) | |
| **Overall satisfaction** (0 = low, 100 = high) | | | |
| How confident do you feel about recognising that a child's health is deteriorating? | 90.41 (10.44) | 80.72 (16.30) | p = .024 |
| What overall score would you assign VitalPAC in terms of your satisfaction? | 78.97 (17.20) | 55.82 (33.21) | p = .012 |
| How satisfied are you with the ability to obtain a charged hand-held device to perform your observations on VitalPAC? | 2.10 (.94) | 2.71 (1.16) | p = .016 |
| **Education, training and implementation** (1 = high, 5 = low) | | | |
| How satisfied are you with the education and training you received? | 2.00 (1.02) | 2.59 (1.33) | p = .094 |
| How confident do you feel that your education and training on VitalPAC permit you to respond effectively to acutely ill patients? | 2.01 (.92) | 2.35 (1.46) | p = .353 |
| How satisfied are you with the way VitalPAC is implemented in your area? | 1.86 (.82) | 2.94 (1.25) | p = .003 |
| **Recording and monitoring** (1 = high, 5 = low) | | | |
| How satisfied are you that VitalPAC allows you to record accurate data? | 2.08 (.93) | 2.59 (1.06) | p = .040 |
| How confident are you in the way in which VitalPAC monitor your patients for deterioration? | 2.00 (.76) | 2.82 (1.33) | p = .023 |
| How satisfied are you that VitalPAC will reduce the incidence of the omission of recording of vital signs? | 2.30 (.92) | 2.71 (1.16) | p = .096 |
| Completeness of documentation | 1.81 (.79) | 2.59 (1.23) | p = .020 |
| Frequency of documentation | 1.91 (.80) | 2.41 (.94) | p = .019 |
| **Recognition, awareness and level of concern** (1 = high, 5 = low) | | | |
| How confident are you that VitalPAC escalation reflects the clinical decision you want to make? | 2.23 (.83) | 3.06 (1.30) | p = .001 |
| How satisfied are you with the way in which VitalPAC supports you in recognising deterioration? | 1.95 (.76) | 2.82 (1.33) | p = .017 |
| How confident are you that VitalPAC reflects your level of concern? | 2.14 (.85) | 2.88 (1.22) | p = .002 |
| How confident are you that VitalPAC helps make you aware of the sickest children in your setting/area of responsibility? | 2.10 (.84) | 2.76 (1.35) | p = .065 |
| Real time oversight of the sickest patients | 1.95 (.82) | 2.71 (1.31) | p = .033 |
| How satisfied are you that VitalPAC allows you to visualise trends in data efficiently? | 2.16 (.90) | 2.75 (1.53) | p = .151 |
| **Escalation, decision making and timeliness of response** (1 = high, 5 = low) | | | |
| How confident are you that VitalPAC ensures that patients who require escalation are promptly referred to the appropriate clinician? | 2.20 (.94) | 3.12 (1.22) | p = .001 |
| How confident are you that VitalPAC assists a timely response to signs of deterioration? | 2.01 (.89) | 2.76 (1.52) | p = .061 |
| **Usability** (1 = high, 5 = low) | | | |
| Ease of use | 1.65 (.90) | 2.50 (1.27) | p = .001 |
| View of completed observations | 1.95 (1.01) | 2.73 (1.49) | p = .065 |
| Careflow Connect | 2.11 (1.12) | 2.44 (1.67) | p = .463 |
| Availability of devices | 1.93 (.94) | 2.44 (1.37) | p = .164 |
| Speed of data | 1.98 (.98) | 2.50 (1.16) | p = .050 |
| Icons | 1.90 (.80) | 2.63 (1.09) | p = .020 |
| Automated prompts | 1.88 (.88) | 2.69 (1.30) | p = .027 |
| Automated doctor alert system | 1.94 (.96) | 2.81 (1.38) | p = .025 |

\* **Note**: At start of the study Vitals and Connect CareFlow (DETECT e-PEWS) were called VitalPAC. T tests were conducted to compare the D-VS and R-VS groups on the continuous variables. Means, SD, and p values are reports in the table and statistically significant t tests are reported in the text.

was not statistically significant. However, some open responses suggested that, despite training, staff did not always directly record vital signs in real-time:

*sometimes you do observations then have to do other cares and forget to record on DETECT [and] it's still on a piece of paper (Staff Nurse, D-VS, 150).*

Satisfaction with DETECT e-PEWS was significantly higher for the D-VS group compared to the R-VS group in terms of both 'completeness of documentation' ($t$ (17.88) = -2.55, $p$ = .020) and 'frequency of documentation' ($t$ (139) = -2.38, $p$ = .019).

**Recognition, awareness and level of concern.** The D-VS group had significantly higher levels of 'confidence in the way DETECT e-PEWS supports recognition of deterioration' than the R-VS group ($t$ (17.45) = -2.64, $p$ = .017). A typical positive open response noted that DETECT e-PEWS:

*allows you to see trends in previous PEWS recorded and alerts you if there are any concerns if the PEWS are out of normal limits (Staff Nurse, D-VS, 88).*

The D-VS group also had more confidence in DETECT e-PEWS than the R-VS group when it came to the extent to which the device 'reflects your level of concern' and this difference between groups was statistically significant ($t$ (139) = -3.20, $p$ = .002). One member of staff noted that it was *"good that it captures parental concern"* (Advanced Practitioner, R-VS, 23). Only one participant noted that *"there have been occasions where I have been more concerned than reflected on system"* (Sister, D-VS, 141).

In terms of the extent to which the device helped raise awareness of the sickest children in the setting/area of responsibility', confidence was once again higher amongst the D-VS group, although this difference only approached statistical significance ($t$ (17.76) = -2.31, $p$ = .033). However, some participants rejected DETECT e-PEWS' contributions noting *"it doesn't make a difference. We know who our most unwell patient is without it"* (Staff Nurse, D-VS, 52).

In terms of 'real time oversight of the sickest patients'; confidence was once again significantly higher in the D-VS group than the R-VS group ($t$ (17.76) = -2.31, $p$ = .033). However, some staff raised concerns about alerts being triggered when children's baseline (e.g., cardiac or complex healthcare needs) vital signs are outside of the standard limits, for example:

*some of our complex patients trigger high PEWs even when well and may not be the sickest patient on the ward (Sister, D-VS, 147).*

There was no difference between groups as to the extent that DETECT e-PEWS 'allows professionals to visualise trends efficiently'. Although most staff were satisfied with the trends and liked *"being able to see graphs as it shows trends easily"* (Advanced Practitioner, R-VS, 16). Respondents differed in opinions about whether DETECT e-PEWS provided better visualisations than Meditech: one participant noted that they preferred visualising trends on DETECT e-PEWS as *"vital signs and pew are graphically displayed is much better than on Meditech"* (Doctor, R-VS, 19) whereas another preferred Meditech as the *"screen [is] larger. . . more data"* (Staff Nurse, D-VS, 52).

**Escalation, decision making and timeliness of response.** The D-VS group had significantly higher 'confidence that patients requiring escalation of care are promptly referred to the appropriate health professional' than the R-VS group ($t$ (137) = -3.62, $p$ = .001).

The D-VS group had higher 'confidence that Vitals [e-PEW score app] assists a timely response to signs of deterioration' than the R-VS group although this group comparison was not statistically significant. Some concern was raised in the open text responses such as being unsure about whether *"doctors always receive messages, end up bleeping on phone"* (Staff Nurse,

D-VS, 94) or *"this system does not alert you if you are busy with another patient in the way that a bleep does and this can result in a delay"* (Doctor, R-VS, 143).

However, positive responses were typified by the visual cues and how it could:

*support you to demonstrate escalation is required, by showing an upward or downward trend, whichever is relevant (Advanced Practitioner, R-VS, 16).*

However, it was also noted that staff would "also use my own assessment" (Sister D-VS, 32).

**Usability.** Overall, usability was high. The D-VS group had higher levels of satisfaction in terms of 'ease of use' (usability) compared to the R-VS group and this difference was statistically significant ($t$ (138) = -3.39, $p$ = .001). Although most open responses about usability were positive, some negative responses reflected the following concerns such as *"having multiple places to record information is confusing and complicated"* (Staff Nurse, D-VS, 56) and reviewing vital signs is *"no different to Meditech [and] much harder to see on smaller screens such as ipad"* (Doctor, R-VS, 139).

There was concern raised about there being *"too many devices and means of communicating [in the hospital] already"* (Doctor, R-VS, 18). Satisfaction with DETECT e-PEWS also reflected how embedded it was on a particular ward with staff in some settings seeing it as less suitable for their setting, for example, *"designed to be more ward based. . .not HDU specific"* (Staff Nurse, D-VS, 33).

The D-VS and R-VS groups reported lower levels of satisfaction in relation to CareFlow Connect [response app] compared to other usability characteristics. Open text responses showed it was not used consistently across all settings, such as *"CareFlow is not commonly used by MDT"* (Staff Nurse, D-VS, 98) and not always thought to *"make my job easier"* (Doctor, R-VS, 139).

Although both groups were similar in their satisfaction regarding the availability of devices (iPods or iPads), satisfaction was significantly higher in the D-VS group than the R-VS group in terms of speed of data input ($t$ (137) = 1.97, $p$ = .050), icons ($t$ (17.18) = -2.57, $p$ = .020), automated prompts ($t$ (16.84) = -2.42, $p$ = .027), and automated doctor alert system ($t$ (16.95) = -2.45, $p$ = .025).

## Discussion

This is the first paper describing the clinical utility and acceptability of a hospital-wide, proactive end-to-end deterioration solution (the DETECT surveillance system) with an embedded e-PEWS that included sepsis screening. The DETECT surveillance system aims to proactively screen paediatric patients for early signs of serious deterioration or sepsis, create alerts, and escalate concerns to reduce complications and emergency transfers to critical care following deterioration in hospital.

The discussion contextualises the perceptions of the clinical utility [33] and acceptability in line with our stated definitions of these concepts [37]. However, we frame the discussion within the five rings of the chain of prevention (Fig 1) [21] and we note that whilst Smith's focus is entirely on health professionals, ours encompasses parents. We chose to structure the findings using the chain of prevention as each 'ring' is a discrete component important in the prevention of deterioration. When specifically considered, the acceptability constructs from Theoretical Framework of Acceptability v2 [37] (see also Fig 3) are signposted in brackets as Construct 1, Construct 2 etc. As seen in other PEWS studies, implementation is challenging and system-wide changes need organisational support [43].

## Education

Overall, the clinical utility of the training was good and acceptability was good in that professionals felt satisfied, confident, well prepared, and able to respond effectively to acutely ill children. Although the chain of prevention focuses on education of staff [21], it was interesting to note that the implementation of the system created opportunities for professionals to explain DETECT e-PEWS and the devices used, talk about vital signs, and for parents to ask questions about the technology; thus supporting attainment of 'intervention coherence'. This serendipitous parent training may prove beneficial, as studies addressing parent involvement in the escalation of care note that some professionals doubt parent capabilities [44] and have concerns about misuse of escalation [45].

Health professionals were supported by their initial and ongoing training and education promoting a sense of 'self-efficacy' (Construct 7). Success is known to be supported by factors including education which addresses the value of technology or the intervention [34], makes staff curious [46] and which enhances 'affective attitudes' (Construct 1) [37, 46]. Education is key to understanding processes ('intervention coherence') (Construct 4), and in the DETECT study both implementation and assimilation were ongoing processes, as recommended as this is known to be core to changing practice [35, 47].

## Monitoring

The clinical utility of DETECT e-PEWS in terms of its ease of use in recording of vital signs via the app on the iPods was considered good by most health professionals. Generally, PEWS studies only consider monitoring acceptability from the perspective of health professionals [6, 7, 13]; however, our study also addressed acceptability from the perspectives of parents. Parents trusted DETECT e-PEWS, as they believed that it was efficient, better than 'just paper', and made them feel safe and it demonstrated robust acceptability across all aspects of acceptability (Constructs 1–7). However, acceptability could have been improved for some parents if more information (e.g., the results of their child's vital signs) had been shared with them. It is interesting to note that other escalation of care studies focus attention on information and/or education about how to express concern [48–50], but do not present evidence of educating parents about their child's vital signs.

Acceptability was good overall for health professionals with most preferring the DETECT e-PEWS over paper-based scoring in terms of, for example, its 'perceived effectiveness' (Construct 6) (e.g., in reducing workload), its interface, icons, automated prompts and how it supported completeness of documentation. Such factors are key to the successful implementation of digital health interventions [46, 51]. Acceptability was good in terms of ethicality (Construct 3) as the DETECT system fitted with the 'values, priorities and routines' [52] particularly of the D-VS group who absorbed any 'opportunity costs' (Construct 5) into their everyday practice, and demonstrated clear 'self-efficacy'(Construct 7) [37] in their confident engagement with the DETECT system.

## Recognition

Most health professionals had confidence (better in D-VS than R-VS group) in the clinical utility of DETECT e-PEWS in triggering recognition of potential deterioration. Some health professionals in speciality settings (e.g., cardiac care and high dependency) identified that the predefined alert scores were inappropriately sensitive in triggering alerts.

Parents' perception of the automated calculation of scores component of DETECT e-PEWS reflected high acceptability ('perceived effectiveness') (Construct 6) as it would 'keep their child safe' and because it included parental concern.

Overall, acceptability was good (higher in D-VS than R-VS group), with most health professionals seeing benefits ('ethicality' and 'affective attitude') (Constructs 1 and 3) such as liking the real-time and/or remote visualisation of trends and as seen in other studies [14, 53]. Most health professionals trusted the DETECT system ('perceived effectiveness', Construct 6) to better support recognition of deterioration, a core aspect inherent in the chain of prevention [21], further reflecting the 'ethicality' (Construct 3) of acceptability. However, as with other studies of digital health implementation, some health professionals were reticent, perhaps seeing the 'opportunity costs' (Construct 5) outweighing benefits, as they questioned the need for automation and/or considered the DETECT system a threat to their clinical judgement as seen in other work [53]. Clearly opportunity costs do need better consideration in future implementation work and attention needs to be paid to how perceived threats can be better managed.

## The call for help

Although the DETECT system's clinical utility was generally high in relation to automated alerts there were some concerns that the system might be less effective than 'bleeping' (paging) a doctor, as some health professionals were unsure if triggered messages were received. Lack of certainty and concerns about variation in responsiveness have been shown to be barriers [53]. The clinical utility of the DETECT system depends on its accuracy in supporting health professionals across general and speciality settings and avoiding problems such as 'call fatigue' [53], which has been reported as a barrier when alerts are triggered inappropriately.

Parents had positive 'affective attitudes' (Construct 1) [37] toward the DETECT system, knowing that it would trigger an auto alert and 'call for help' without relying on a health professional to make the call. Most parents reported that they know when their child is 'getting poorlier', but it is unclear from the survey how confident parents felt in voicing these concerns, or how comfortable they felt in responding to the health professional asking them the 'parental concern' question as part of doing their vital signs. Other studies have shown that some parents lack confidence in raising and/or escalating concerns [44, 49] or challenging medical staff [22] and that concerns raised by relatives are not always related to deterioration [45, 54]. This occurs despite endorsement of national and international bodies in promoting consumer voices in escalation [45].

Overall, health professionals had positive 'affective attitudes' (Construct 1) to the DETECT system reflecting its acceptability. However, the response component (CareFlow Connect app) of the DETECT system had yet to reach similar levels of acceptability in some sub-sets of the R-VS group, perhaps reflecting that this group were more aware of 'opportunity costs' (Construct 5) [37] as they were less convinced by the net benefit [55], and value [34] which may have led to low levels of social proof (recommendation by peers) [32]. 'Burden' (Construct 2) of use can reduce acceptability [37] and the main complaint with some members in the R-VS group arose from the need to carry an additional piece of technology (iPods) with them. The requirement for apps to be device agnostic would help reduce the number of devices being carried and could reduce the burden.

## Response

Overall, most health professionals had confidence in the clinical utility of the DETECT system in relation to response, although this was better in the D-VS than R-VS. group. Parents who had experienced a CDE whilst the system was in place reported high acceptability reflecting its 'perceived effectiveness' (Construct 6) for their children's safety.

Overall, health professionals had positive 'affective attitudes' and positive comments about the response component of the DETECT system, such as access to real-time data [53].

However, the CareFlow Connect app had been in place only a few weeks before the first lockdown of the COVID-19 pandemic. While its use was recommended as part of the DETECT study, this was difficult to mandate because clinical teams had to adapt quickly to work differently to address challenges associated with staffing, cross cover of patients and other challenges. Some scepticism about the response component of DETECT e-PEWS, held by some of the R-VS group, may reflect negative affect in relation to fears of the 'burden' (Construct 2) associated with suspected hidden work and concerns about the DETECT system not fitting in with their routines and practices ('ethicality', Construct 3), as seen in other e-implementation work [52]. Other studies addressing assimilation of new technologies note that professionalism can be a barrier to smooth implementation. Barriers can be raised as a result of different perspectives held by different professional groups [35]; perceptions of opportunity costs (Construct 5) could be reduced if respected professional champions were given time, support and organisational backing to drive forward implementation.

## Limitations

No specific measures of or cut-offs for utility or acceptability were used, although the DETECT study did use rating scales with open text boxes as advised [32]. The lack of validated measures for the concepts of interest can be seen to be a limitation. Various factors limit the samples of parents and health professionals and thus potentially limit the validity and robustness of the findings. One key limitation that a non-probability sampling technique was used; the limitations associated with convenience sampling include sampling and selection bias, limits to generalisability of findings and less granularity of data. Further, the sample size for parents and professionals is relatively small compared to the population of all parents whose children were receiving care and all professionals using the DETECT system. However, although the HP population does include diverse representation across professions and grades, the findings are significantly more weighted to professionals in the D-VS group than the R-VS group. Although two settings (Cardiac and HDU) were less represented, their staff would have had similar access to DETECT system as other areas. This lower representation may be linked to the constraints related to COVID-19 measures created more limited access to these settings for data collection. The sample of parents is not likely to be as diverse as the whole population of eligible parents; a more targeted matrix sampling approach might be considered in future. Additionally, recruitment of parents occurred during the Covid-19 pandemic (fewer admissions) and we were not able to recruit consistently across all months that the study was open due to staff shortages, reduced access to wards). Thus, the population of non-CDE children may not be representative of the total hospital population pre-pandemic (e.g., elective surgeries cancelled, only the acutely unwell children remained or were admitted to hospital). However, our pre-pandemic baseline data (not reported in this paper) suggests that our CDE population is representative as pre-pandemic critical deterioration occurred, most commonly, in children who were acutely unwell or required emergency surgical care.

The challenge of implementing the response component (CareFlow Connect app) of the DETECT system within a hospital under extraordinary pressure from the impact of COVID-19 limits what can be stated about this aspect of the system. These limitations mean that the generalisability of the results is limited.

## Conclusion

Overall clinical utility and acceptability were positive, although there was evidence that liking/satisfaction dropped over time; as with most implementation strategies, assimilation is an ongoing process [35] requiring effort to sustain both motivation and a sense of positivity

across the TFA's constructs [37]. However, acceptability was evident across all seven constructs. Considering the multifaceted nature of the intervention and the complexity of the implementation across a whole hospital as part of a research study, rather than an organisationally driven programme, it is evident that the DETECT system has had success across two key groups of stakeholders: parents and health professionals. As the DETECT system is handed over to the organisation for ongoing embedding, the findings from the survey when considered in relation to both the chain of prevention [21] and the TFA, provide clear indications as to where the links in the chain need strengthening and where effort is required to enhance acceptability.

## Supporting information

**S1 Checklist. Checklist for Reporting Of Survey Studies (CROSS).**
(DOCX)

## Acknowledgments

We wish to acknowledge the parents and health professionals who participated in the survey and the research nurses who supported the collection of data during the COVID-19 lockdowns.

## Author Contributions

**Conceptualization:** Bernie Carter, Matthew Peak, Gerri Sefton, Enitan D. Carrol.

**Data curation:** Gerri Sefton, Enitan D. Carrol.

**Formal analysis:** Bernie Carter, Holly Saron, Lucy Blake, Jane Harris.

**Funding acquisition:** Bernie Carter, Matthew Peak, Gerri Sefton, Enitan D. Carrol.

**Investigation:** Bernie Carter, Holly Saron.

**Methodology:** Bernie Carter, Matthew Peak, Gerri Sefton, Enitan D. Carrol.

**Project administration:** Caroline Lambert.

**Visualization:** Bernie Carter, Lucy Blake.

**Writing – original draft:** Bernie Carter, Holly Saron, Lucy Blake.

**Writing – review & editing:** Bernie Carter, Holly Saron, Lucy Blake, Chin-Kien Eyton-Chong, Sarah Dee, Leah Evans, Jane Harris, Hannah Hughes, Dawn Jones, Caroline Lambert, Steven Lane, Fulya Mehta, Matthew Peak, Jennifer Preston, Sarah Siner, Gerri Sefton, Enitan D. Carrol.

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
