## [Decision Letter · Decision Letter 0]

24 Nov 2021

PONE-D-21-34519Clinical utility and acceptability of a whole-hospital, pro-active electronic paediatric early warning system (the DETECT study): a prospective e-survey of children, parents and health professionals.PLOS ONE

Dear Dr. Carter,

Thank you for submitting your manuscript to PLOS ONE. After careful consideration, we feel that it has merit but does not fully meet PLOS ONE’s publication criteria as it currently stands. Therefore, we invite you to submit a revised version of the manuscript that addresses the points raised during the review process.

We look forward to receiving your revised manuscript.

Kind regards,

Jagan Kumar Baskaradoss

Academic Editor

PLOS ONE

Journal Requirements:

Reviewers' comments:

Reviewer's Responses to Questions

**Comments to the Author**

1. Is the manuscript technically sound, and do the data support the conclusions?

Reviewer #1: Partly

Reviewer #2: No

2. Has the statistical analysis been performed appropriately and rigorously? 

Reviewer #1: Yes

Reviewer #2: No

3. Have the authors made all data underlying the findings in their manuscript fully available?

Reviewer #1: No

Reviewer #2: No

4. Is the manuscript presented in an intelligible fashion and written in standard English?

Reviewer #1: Yes

Reviewer #2: Yes

5. Review Comments to the Author

Reviewer #1: This paper reports on surveys exploring clinical utility and acceptability of an

electronic paediatric early warning system from the perspective of parents, children and health professionals as a component of a larger implementation study. The study findings will be of interest after the manuscript is further developed. Following and citing a reporting checklist such CROSS as will strengthen the manuscript https://www.equator-network.org/?post_type=eq_guidelines&eq_guidelines_study_design=0&eq_guidelines_clinical_specialty=0&eq_guidelines_report_section=0&s=survey&btn_submit=Search+Reporting+Guidelines

P4

Training for health professional is described – but not how patients and families were prepared and supported or aware of the DETECT System

e-handover function – not explained or if used – if not – is it necessary to describe?

final para – need to explain on-call teams and Acute Care Team for international audience

Explain significance of app on own device

The concepts clinical utility and acceptability need to be explained and in the survey development p6 then relate to measures and items that address the concepts of interest

P5

Design is a survey with quantitative and qualitative components. Following a reporting checklist will assist in addressing many of the comments listed.

Participants and setting: need to explain critical deterioration event

The groups of participants is not clear – for Group 1 - perhaps an error of children rather than parents and Group 2 should be parents of children also?

Had the children experienced a CDE?

Did parents provide consent for child’s participation? Was 7 years the minimum age to participate? Was there a rationale for the minimum age?

Was the sample size estimated?

Was recruitment purposive for ethnocultural diversity? Were interpreter services available for participants who did not understand English

Health Professional recruitment needs a bit more explanation – was email to all eligible first and followed up by face to face requests? How was the possibility of coercion managed?

Who is included? Is it nurses and doctors and allied health staff? I can see this reported in results but needs to be defined in methods

P6 The surveys were developed for the purpose of this study and described as non-validated.

There is no description of the measures for clinical utility and acceptability despite these concepts being previously reported by others

There is some description about consulting with parents and children in designing the survey.

There is no description about the development of the health professionals survey. Was there any consultation/involvement with health professionals?

Was there any content validity testing?

How long did the survey take to complete? Did the researcher assist families to complete?

P7 Ethics

Not described is how participants received information about the study results

Results

These are participant characteristics not demographics – need to change the heading

There is no description of the denominator to understand the response rate. How many potential participants received the survey or how many were requested to complete the survey and declined?

How many completed electronically and how many paper based?

Table 1 – it will be helpful to understand the patients in the high dependency unit – are these patients who have been in PICU?

For international audience the terminology of Assistant Nurse Practitioner, Assistant Practitioner, Advanced Clinical Practitioner and Acute Care Team needs to be explained

Table 2 What are the professions of allied health – Physiotherapist, Occupational Therapist, Pharmacist?

P 10 core findings will be improved by presenting the positive findings first

Overall satisfaction and competence – not clear what the competence relates to. This scale is not described on P6

Qualitative findings:– it is not reported how many parents provided comments nor is it evident whether the quotes are selected from a few or many parents’ comments.

P12 Findings from health professionals’ surveys – need to report the actual findings and statistics intext and it will improve the readability to report what was found first then detail differences. The reporting using headings linked to Smith’s chain of prevention should be described if this was planned

This section of the manuscript p12 – 14 needs the most work as it is hard to follow.

The last few lines p12 and on p15 there are statistics provided but these are not clearly presented

No satisfaction scale described in survey development but reported here

Were there differences in responses based on profession or professional experience?

P15 “A similar pattern…. “ this need to be reworded to explain the finding first

Discussion

The concepts of clinical utility and acceptability are raised here but there needs to be greater clarity informing the survey. This section is insufficiently developed and is difficult to follow

The discussion should more clearly identify how this study adds to or confirms or refutes others’ research in the area and include recommendations

Limitations

The lack of measures for the concepts of interest is a major limitation

The small sample of health professionals is acknowledged but the sample of 137 parents and sample of 8 children is not acknowledged

Generalisability should be addressed

Conclusion

This should be stand alone and highlight key findings ie not refer to figure

Reviewer #2: General Comments:

This study aimed to examine how parents, children and health professionals view and engage with the DETECT electronic Paediatric early warning systems (PEWS) apps, with a particular focus on its clinical utility and its acceptability.

Overall, the study is well-written and presents interesting and novel findings. I have some major and some minor comments, which needs to be addressed before proceeding further.

Major Comments:

The study employed a non-probability sampling technique for selection of samples. This method has several limitations and could limit the validity of the study results. The authors have not discussed this issue.

Recruitment of participants was done during the ongoing pandemic. This could influence the characteristics of patients included in the study. They may not be representative of the patients attending the hospital prior to the pandemic. This has to be discussed.

Was any power analysis done? How did the authors decide on the sample size requirement?

I believe the category of children is severely under powered to derive any meaningful conclusions. I suggest the authors add more children to the sample or eliminate this group from analysis.

Was any piloting of the questionnaire performed?

“parents/carers were engaged with via two face-to-face workshop groups (n=8) and by email (n=3) ” these are two different techniques, which can influence the validity of the results.

The analysis is incomplete. I recommend that the authors take the help of an experienced statistician to enhance the data analysis.

Minor Comments:

• The referencing style in not in accordance with the journal’s style. Please review the author instructions or refer to any recent paper published in the Journal.

• Abstract; open and closed question? Clarify..

• Materials and Methods:

o Prospective or cross-sectional?

o young people (aged 7-18 years old) ? adolescents?

o Group 1 (children whose children had not experienced a critical deterioration event during admission…) ? Revise

o “Although consent is not required for NHS professionals involved in evaluating an intervention, consent from the health professionals was gained via a ‘tick box’ on the survey. “ Incorrect statement. Consent is implied for procedures involving diagnosis or treatments withing the hospital facilities. This was a research project were a new instrument was being investigated. Any research involving human subjects require ethical approval (Declaration of Helsinki).

o Analysis: inputted?

o Mean and SD are descriptive statistics. How can this be used to compare distributions? List any statistical test used..

• nfe

6. PLOS authors have the option to publish the peer review history of their article (what does this mean?). If published, this will include your full peer review and any attached files.

Reviewer #1: **Yes: **Associate Professor Fenella J Gill

Reviewer #2: No

---

## [Author Response · Author response to Decision Letter 0]

13 Feb 2022

Please also see the word document which may be easier to read (as it includes additional text in red font)!

Rebuttal to reviewers.

Thank you both for giving up your time to review our paper and to provide such constructive comments.

We have been conscientious in our response to each of your comments and we know believe that this version is stronger and more robust.

All new text is presented in red.

Review Comments to the Author

Reviewer #1: 

Thank you for your helpful and insightful comments on our paper. We’re grateful for the time you’ve take to help us improve the manuscript. We also appreciate the time it will take to consider our revisions.

This paper reports on surveys exploring clinical utility and acceptability of an

electronic paediatric early warning system from the perspective of parents, children and health professionals as a component of a larger implementation study. The study findings will be of interest after the manuscript is further developed. Following and citing a reporting checklist such CROSS as will strengthen the manuscript https://www.equator-network.org/?post_type=eq_guidelines&eq_guidelines_study_design=0&eq_guidelines_clinical_specialty=0&eq_guidelines_report_section=0&s=survey&btn_submit=Search+Reporting+Guidelines

Thank you for this comment, we have used the CROSS checklist (Sharma et al, 2021) as suggested.

(p5) The Consensus-based checklist for reporting survey studies (CROSS) [38] has been used to ensure high quality reporting.

P4 Training for health professional is described – but not how patients and families were prepared and supported or aware of the DETECT System. 

(p4) Children and parents were made aware of the implementation of the technology through the use of dedicated posters in all public areas and an explanation provided at the child’s admission.

e-handover function – not explained or if used – if not – is it necessary to describe? 

Since we do not mention e-handover within the results or discussion we have removed mention of it.

final para – need to explain on-call teams and Acute Care Team for international audience. 

(pp4-5) The clinical teams had a minimum of one iPod per team; each member of the on-call team (medical team that provides out-of-hours cross cover for inpatients and new admissions) and those within the Acute Care Team (nurse-led Rapid Response Team) each had an iPod or iPad (some used their personal mobile phone).

Explain significance of app on own device 

(p5) Additionally, there was an agreement that staff could have the Careflow Vitals and Connect apps loaded to their personal device under the ‘Bring Your Own Device’ scheme which meant that some staff did not have to carry an additional device and it was more convenient for them (Vitals is device specific (Apple), Connect is device agnostic and works on all personal devices).

The concepts clinical utility and acceptability need to be explained 

The definitions of clinical utility and acceptability previously presented in the Discussion section have been moved forward to the Introduction section, see below.

(p5) Defining clinical utility and acceptability

It is important to define the concepts of clinical utility and acceptability, as it is evident in the literature that there is ambiguity and overlap in what is encompassed by the terms, and there is no consensus on definitions 32. Within this discussion, clinical utility is defined in its narrowest sense; does the technology do what it is supposed to do, and does it perform its designated function 33? 

However, the complexity inherent in implementation, adoption and assimilation of technology in healthcare systems 34 35 requires the definition of acceptability to acknowledge its multifaceted nature, and to be more encompassing 32 34 36 37. The Theoretical Framework of Acceptability (TFA) (v2) 37 is composed of seven component constructs (Figure 3): ‘affective attitude’, ‘burden’, ‘ethicality’, ‘intervention coherence’, ‘opportunity costs’, ‘perceived effectiveness’ and ‘self-efficacy’. The TFA proposes that acceptability is a “multi-faceted construct that reflects the extent to which people delivering or receiving a healthcare intervention consider it to be appropriate, based on anticipated or experienced cognitive and emotional responses to the intervention” 37. 

And in the survey development p6 then relate to measures and items that address the concepts of interest…

Note: the surveys were designed to ask broad questions of interest rather than specifically address seven component constructs of the TFA. We have made this clearer in the parent survey section 

(p7) …….. (4) About the Device (n=9 questions asking about satisfaction with explanations about the device and how it was used, trust in technology, feeling safe and secure with the device (relating to our definitions of utility 33 and acceptability 37). 

And in the HP survey section we used the domains from Smith’s Chain of Prevention to structure the survey as this structure was familiar to staff, although the concepts of the TFA were embedded:

(p7) The health professionals’ survey consisted of eight sections, five of which (education, monitoring, recognition, escalation, and response) related to the chain of prevention 21: Introduction; Deciding to Take Part; Education and Training, (n=2 questions); Monitoring and Recognition (n=5 questions); Escalation and Response (n=3 questions); System Features (n= 4 questions), Concluding Thoughts (n=3 questions); all but the first section related to our definitions of utility 33 and acceptability 37.

P5

Design is a survey with quantitative and qualitative components. Following a reporting checklist will assist in addressing many of the comments listed.

(p5) Thank you for this suggestion we have adopted the CROSS checklist (Sharma et al., 2021)

Participants and setting: need to explain critical deterioration event 

We have added in the definition of a CDE as follows:

(p6) A CDE was defined as a deterioration where the patient is critically unwell, which culminates in an emergency transfer to high dependency unit or the intensive care unit, or an unexpected death.

The groups of participants is not clear – for Group 1 - perhaps an error of children rather than parents and Group 2 should be parents of children also? 

(p6) This error has been corrected.

Had the children experienced a CDE?

(p6) Yes, children in Group 2 had experienced a CDE.

Did parents provide consent for child’s participation? Was 7 years the minimum age to participate? Was there a rationale for the minimum age?

Note, in response to your questions. Parents did provide consent for their child’s participation, 7 years was the minimum age for the child’s active participation (e.g., completing survey) in the study, and 7 years was deemed a sufficient age to give assent and to be able to complete survey. However, based on Reviewer 2’s comments about the inadequacy of the children’s dataset we have removed reference to children’s data from the paper.

Was the sample size estimated? 

As the surveys were not validated and only intended to generate descriptive data we did not do sample size calculations. Our intention was to gain initial perspectives upon which to base future surveys. Additionally, the survey aimed to provide a baseline for the in-depth qualitative interviews we did as part of the study. 

Was recruitment purposive for ethnocultural diversity? Were interpreter services available for participants who did not understand English.

Recruitment of parents was opportunistic and not purposive for diversity. Interpreter services were not available.

(p7) The surveys were only available in English and no dedicated translation was available.

Health Professional recruitment needs a bit more explanation – was email to all eligible first and followed up by face to face requests? How was the possibility of coercion managed?

(p6) Recruitment of health professionals (doctors, nurses and allied health professionals, n=151) with experience of using the system was either by email or opportunistically face-to-face on the wards by the researcher asking staff if they were interested. Health professionals were also given detailed information via information sheets, given sufficient time to consider if they wanted to participate and the opportunity to ask questions (face-to-face or remotely).The possibility of coercion was avoided by making it clear that participation was voluntary and leaving the device with the survey link on it with the potential participant for about fifteen minutes; allowing the participant to complete the survey or not, as preferred.

Who is included? Is it nurses and doctors and allied health staff? I can see this reported in results but needs to be defined in methods

This has been corrected (see also above):

(p6) Recruitment of health professionals (doctors, nurses and allied health professionals, n=151) with experience of using the system …

P6 The surveys were developed for the purpose of this study and described as non-validated.

There is no description of the measures for clinical utility and acceptability despite these concepts being previously reported by others

As noted previously, we have moved the description of the definitions of clinical utility an acceptability into the Introduction section (p5).

There is some description about consulting with parents and children in designing the survey.

There is no description about the development of the health professionals survey. Was there any consultation/involvement with health professionals? 

We did consult with health professionals. We have added this in.

(p7) Consultation with health professionals (paediatric doctors and nurses who were part of our wider steering group) helped to develop the structure, content and readability of the health professional survey

Was there any content validity testing?

We did not do formal content validity testing although we did do some pretesting, see below:

(p7) Pretesting/piloting of our proposed final versions of the surveys was carried out with parents (see engagement in previous section) and health professionals (nurses and doctors, n=5) was carried out on one occasion; no revisions were identified as being required.

How long did the survey take to complete? 

(p7) Three versions, each taking about 3-5 minutes to complete, were created one for health professionals…

Did the researcher assist families to complete? 

We have addressed this comment in text.

(p7) Typically, the researcher did not assist parents to complete the survey, although support was available, as needed. 

P7 Ethics

Not described is how participants received information about the study results. 

We have addressed this in text.

(p8) Direct feedback to individual survey participants was not possible (due to anonymity of survey responses) but findings will be shared with the broad population of parents through the Parent and Carer’s Research Forum, hospital newsletter, social media etc. and with health professionals via Grand Rounds and other meetings. 

Results

These are participant characteristics not demographics – need to change the heading. 

(p8) Heading changed as requested.

There is no description of the denominator to understand the response rate. How many potential participants received the survey or how many were requested to complete the survey and declined?

We have added this information in:

(p8) Of the parents approached, there was a 9-10% decline rate (typical reasons for declining being focused on child). In total, 137 parents completed the survey….

(p10) In total 151 health professionals participated in the survey (decline rate not calculated as staff were approached by email as well as directly but typical reason for declining being ‘too busy’); …….

How many completed electronically and how many paper based?

We have added this information in. 

(p8) Parents: All but three surveys (n=134) were completed electronically.

(p10) HPs: All completed electronically.

Table 1 – it will be helpful to understand the patients in the high dependency unit – are these patients who have been in PICU?

(p9) Table 1 has been amended with a note to describe HDU occupancy.

* The high dependency unit (HDU) provides level 2 critical care 39. The HDU patient population includes patients who have deteriorated on the ward, high acuity patients post-operatively as well as some step-downs from PICU with higher care needs than can be delivered on a ward.

For international audience the terminology of Assistant Nurse Practitioner, Assistant Practitioner, Advanced Clinical Practitioner and Acute Care Team needs to be explained.

(p10) We have added this information as a note into Table 2:

* Allied Health Professional is a term that includes physiotherapists and occupational therapists. We did not collect data on the specific profession of AHPs.

**Assistant Practitioners are not registered practitioners but they support care and have a high level of skill through their experience and training 40.

*** Advanced Clinical Practitioners are nurses or AHPs trained to Masters level on an approved ACP course who deliver clinical caseload management autonomously to acute and complex patient groups 41.

**** Acute Care Team is the nurse led Rapid Response Team in the study hospital.

Table 2 What are the professions of allied health 

(p10) See comment above, we have added this information into the note at the bottom of Table 2.

P 10 core findings will be improved by presenting the positive findings first.

Where this has been possible to do/made sense in terms of the ‘narrative’, we have presented the positive findings first in each sub-theme.

Overall satisfaction and competence – not clear what the competence relates to. This scale is not described on P6. 

This was an error on our part, reference to competence removed.

Qualitative findings:– it is not reported how many parents provided comments x

We have addressed this for both parents and HPs.

(p8) Around half the parents (n= 27 CDE and n=38 non-CDE) provided open text responses. 

(p10) Forty four percent (n=66) of HPs provided at least one open text, with just under half of these (n=25) at least three open text responses; some provided up to nine.

Nor is it evident whether the quotes are selected from a few or many parents’ comments. 

Within the limited word limit we had drawn quotations from across the CDE and non-CDE parents and across the HPs. This is now more apparent as we have now added labels to all quotations, see an example from parents’ findings:

(p12) One parent noted that the “device seems to make ‘obs’ quicker” (CDE, P46) with another noting it was “wonderful for speed and efficiency…and a great observation checklist for the nurses” (CDE P57). Typical responses included parents thinking that the technology “lowers the risk of mistakes being made when using paper” (CDE P3), delivers the “right results we need to know about her” (CDE P9) and noting that if HPs “do obs on paper they can lose paper obs and have to do them again” (non-CDE P11). 

P12 Findings from health professionals’ surveys – need to report the actual findings and statistics in text and it will improve the readability to report what was found first then detail differences. 

Thank you for your comment here. We have worked with statisticians to enhance the analysis and presentation of the statistics. However, we have retained a view that our intention has primarily been to present descriptive statistics. 

Arising from the discussions we have removed the comparisons in the parent data but retained these within the health professional data.

We have added in more detail about the statistical tests used in the Materials and Methods section (sub-section Analysis):

(p8) Descriptive statistics, mean (M) and standard deviation (SD), are presented to describe variables measured on a continuous scale, categorical variables are reported using counts and percentages. For the health professional data, Chi-squared and Fishers exact test were used to assess between group differences when the outcome of interest was categorical and independent T-test was used when outcome was continuous.

We have added in t tests and p values in the reporting of the HP data, as appropriate, within the main text and provided an explanation of methods at the start of reporting the core findings of the HPs.

(p13) Comparisons were made between groups on the continuous data using t tests. The means, standard deviations and significance levels (p values) are reported in Table 5 and the statistically significant t tests are reported in the text. 

An example of the additions to the main text is provided below.

(p13) However, those in the D-VS group had significantly higher levels of confidence that they could recognise that a child’s health is deteriorating than those in the R-VS group (t (18, 93) = 2.46, p = .024).

Similarly, the D-VS group had significantly higher levels of overall satisfaction with DETECT e-PEWS than those in the R-VS group (t (17,20) = 2.82, p =.012). The D-VS group also had significantly higher levels of satisfaction with being able to ‘obtain a device’ (t (138) = -2.44, p = .016). 

The reporting using headings linked to Smith’s chain of prevention should be described if this was planned. 

We had done this (although perhaps easy to miss) and we direct you to the sentence below.

(p13) First, the data are presented for overall satisfaction and then the remaining results are presented under headings linked to the key aspects of Smith’s [21] chain of prevention.

This section of the manuscript p12 – 14 needs the most work as it is hard to follow. 

We have revised the discussion, and with improved signposting and the additional text that we have included throughout the discussion, this section is now easier to follow. We have presented our work in an important and interesting way. The integrated use of both the Chain of Prevention and the Theoretical Framework of Acceptability is innovative and unlike other published PEWS implementation papers.

We acknowledge that we have used two frameworks to help structure the discussion and this may have been difficult to follow. However, we have now included a new explanatory paragraph and also tried to ensure that the acceptability constructs are better signposted in the text. 

(p14) The discussion contextualises the perceptions of the clinical utility [33] and acceptability in line with our stated definitions of these concepts [37]. However, we frame the discussion within the five rings of the chain of prevention (Figure 1) [21] and we note that whilst Smith’s focus is entirely on health professionals, ours encompasses parents. We chose to structure the findings using the chain of prevention as each ‘ring’ is a discrete component important in the prevention of deterioration. When specifically considered, the acceptability constructs from Theoretical Framework of Acceptability v2 [37] (see also Figure 3) are signposted in brackets as Construct 1, Construct 2 etc.

The last few lines p12 and on p15 there are statistics provided but these are not clearly presented

We have revised our presentation of statistics both in text and within the tables. Additionally, we have removed some comparison data (e.g. influence of length of time using the device) as we felt it did not crucially add to the findings). Note with the changes made to the paper, these no longer appeat on the pages referred to above. 

For example,

(p13) However, those in the D-VS group had significantly higher levels of confidence that they could recognise that a child’s health is deteriorating than those in the R-VS group (t (18, 93) = 2.46, p = .024).

No satisfaction scale described in survey development but reported here. 

We did not use a specific satisfaction scale although specific questions asked participants to indicate their level of satisfaction on a Likert scale in response to an item.

Were there differences in responses based on profession or professional experience?

We did not analyse the data at this level of granularity as we did not believe our sample was large enough.

P15 “A similar pattern…. “ this need to be reworded to explain the finding first. 

The wording ‘similar pattern’ has been removed and the finding reworded, as follows.

(p15) The D-VS group had higher ‘confidence that Vitals [e-PEW score app] assists a timely response to signs of deterioration’ than the R-VS group although the group comparison only approached statistical significance.

Discussion

The concepts of clinical utility and acceptability are raised here but there needs to be greater clarity informing the survey. This section is insufficiently developed and is difficult to follow.

We have removed the some of the description of clinical utility and acceptability to the Introduction section as advised earlier in your comments.

The discussion should more clearly identify how this study adds to or confirms or refutes others’ research in the area and include recommendations

We believe that we have actually already done this in places within the paper as can be seen from the examples below.

• (p17) Generally, PEWS studies only consider monitoring acceptability from the perspective of health professionals [6, 7, 13]; however, our study also addressed acceptability from the perspectives of parents.

• (p17) It is interesting to note that other escalation of care studies focus attention on information and/or education about how to express concern [46-48], but do not present evidence of educating parents about their child’s vital signs. 

However, we have also included some more statements of where we have added to the body of research.

• (p18) Clearly opportunity costs do need better consideration in future implementation work and attention needs to be paid to how perceived threats can be better managed.

• (p18) The requirement for apps to be device agnostic would help reduce the number of devices being carried and could reduce the burden.

• (p19) …………….perceptions of opportunity costs (Construct 5) could be reduced if respected professional champions were given time, support and organisational backing to drive forward implementation.

Limitations

The lack of measures for the concepts of interest is a major limitation

We have now added this to this section.

(p19) The lack of validated measures for the concepts of interest can be seen to be a limitation. 

The small sample of health professionals is acknowledged but the sample of 137 parents and sample of 8 children is not acknowledged.

This has now been addressed.

(p19) The sample size for parents and professionals is relatively small compared to the population of all parents whose children were receiving care

The sample of parents is not likely to be as diverse as the whole population of eligible parents; a more targeted matrix sampling approach might be considered in future.

Generalisability should be addressed.

This has now been addressed.

(p19) These limitations mean that the generalisability of the results is limited.

Conclusion

This should be stand alone and highlight key findings ie not refer to figure. 

We have removed the reference to Figure 3.

Reviewer #2: 

General Comments:

This study aimed to examine how parents, children and health professionals view and engage with the DETECT electronic Paediatric early warning systems (PEWS) apps, with a particular focus on its clinical utility and its acceptability.

Overall, the study is well-written and presents interesting and novel findings. I have some major and some minor comments, which needs to be addressed before proceeding further.

Thank you for your positive comments here and your helpful comments in how to improve the work. We’re grateful for the time you’ve take to help us improve the manuscript. We also appreciate the time it will take to consider our revisions.

Major Comments:

The study employed a non-probability sampling technique for selection of samples. This method has several limitations and could limit the validity of the study results. The authors have not discussed this issue.

We have now added this to the limitations section.

(p19) Various factors limit the samples of parents and health professionals and thus potentially limit the validity and robustness of the findings. One key limitation that a non-probability sampling technique was used; the limitations associated with convenience sampling include sampling and selection bias, limits to generalisability of findings and less granularity of data. Further, the sample size for parents and professionals is relatively small compared to the population of all parents whose children were receiving care and all professionals using the DETECT system.

Recruitment of participants was done during the ongoing pandemic. This could influence the characteristics of patients included in the study. They may not be representative of the patients attending the hospital prior to the pandemic.

We have addressed this below in the limitations section.

(p19) Additionally, recruitment of parents occurred during the Covid-19 pandemic (fewer admissions) and we were not able to recruit consistently across all months that the study was open due to staff shortages, reduced access to wards). Thus, the population of non-CDE children may not be representative of the total hospital population pre-pandemic (e.g., elective surgeries cancelled, only the acutely unwell children remained or were admitted to hospital). However, our pre-pandemic baseline data (not reported in this paper) suggests that our CDE population is representative as pre-pandemic critical deterioration occurred, most commonly, in children who were acutely unwell or required emergency surgical care.

Was any power analysis done? How did the authors decide on the sample size requirement? 

We did not do a power calculation for the sample size. As the surveys were not validated and only intended to generate descriptive data we did not do sample size calculations. 

I believe the category of children is severely under powered to derive any meaningful conclusions. I suggest the authors add more children to the sample or eliminate this group from analysis. 

Based on your suggestion we have eliminated this group from our analysis.

Was any piloting of the questionnaire performed? 

 Yes.

(p7) Pretesting/piloting of our proposed final versions of the surveys was carried out with parents (see engagement in previous section, n=11) and health professionals (nurses and doctors, n=5) was carried out on one occasion; no revisions were identified as being required.

The analysis is incomplete. I recommend that the authors take the help of an experienced statistician to enhance the data analysis. 

Thank you for your comment here. We have worked with statisticians to enhance the analysis and presentation of the statistics. However, we have retained a view that our intention has primarily been to present descriptive statistics. 

Arising from the discussions we have removed the comparisons in the parent data but retained these within the health professional data.

We have added in more detail about the statistical tests used in the Materials and Methods section (sub-section Analysis):

(p8) Descriptive statistics, mean (M) and standard deviation (SD), are presented to describe variables measured on a continuous scale, categorical variables are reported using counts and percentages. For the health professional data, Chi-squared and Fishers exact test were used to assess between group differences when the outcome of interest was categorical and independent T-test was used when outcome was continuous.

We have added in t tests and p values in the reporting of the HP data, as appropriate, within the main text and provided an explanation of methods at the start of reporting the core findings of the HPs.

(p13) Comparisons were made between groups on the continuous data using t tests. The means, standard deviations and significance levels (p values) are reported in Table 5 and the statistically significant t tests are reported in the text. 

An example of the additions to the main text is provided below.

(p13)However, those in the D-VS group had significantly higher levels of confidence that they could recognise that a child’s health is deteriorating than those in the R-VS group (t (18, 93) = 2.46, p = .024).

Similarly, the D-VS group had significantly higher levels of overall satisfaction with DETECT e-PEWS than those in the R-VS group (t (17,20) = 2.82, p =.012). The D-VS group also had significantly higher levels of satisfaction with being able to ‘obtain a device’ (t (138) = -2.44, p = .016). 

Minor Comments:

• The referencing style in not in accordance with the journal’s style. Please review the author instructions or refer to any recent paper published in the Journal.

Referencing style updated to PLoS (as per EndNote)

• Abstract; open and closed question? Clarify..

(p2) closed (tick box or sliding scale) and open (text based) question,

Materials and Methods:

o Prospective or cross-sectional?

(p6) Prospective

o young people (aged 7-18 years old) ? adolescents?

Removed now children/young people/adolescents’ data no longer reported.

o Group 1 (children whose children had not experienced a critical deterioration event during admission…) ? Revise

This has now been revised.

(p6) Group 1 (parents whose children had……

o “Although consent is not required for NHS professionals involved in evaluating an intervention, consent from the health professionals was gained via a ‘tick box’ on the survey. “ Incorrect statement. Consent is implied for procedures involving diagnosis or treatments withing the hospital facilities. This was a research project where a new instrument was being investigated. Any research involving human subjects require ethical approval (Declaration of Helsinki). 

Thank you for this comment. We have deleted the sentence and integrated HPs consent into the same sentence as parental consent. This now reads as follows:

(p6) Consent by parents and health professionals for participation in the survey was gained via a ‘tick box’ at the start of the survey.

o Analysis: inputted?

Inputted deleted.

o Mean and SD are descriptive statistics. How can this be used to compare distributions? List any statistical test used.

Sentence rephrased. 

(p8) Descriptive statistics, mean (M) and standard deviation (SD), are presented to describe variables measured on a continuous scale, categorical variables are reported using counts and percentages. For the health professional data, Chi-squared and Fishers exact test were used to assess between group differences when the outcome of interest was categorical and independent T-test was used when outcome was continuous.

---

## [Decision Letter · Decision Letter 1]

12 Jun 2022

PONE-D-21-34519R1Clinical utility and acceptability of a whole-hospital, pro-active electronic paediatric early warning system (the DETECT study): a prospective e-survey of parents and health professionals.PLOS ONE

Dear Dr. Carter,

Thank you for submitting your manuscript to PLOS ONE. After careful consideration, we feel that it has merit but does not fully meet PLOS ONE’s publication criteria as it currently stands. Therefore, we invite you to submit a revised version of the manuscript that addresses the points raised during the review process.

Please see one minor remaining comment from one of the reviewers below.

We look forward to receiving your revised manuscript.

Kind regards,

Hanna Landenmark

Staff Editor

PLOS ONE

Journal Requirements:

Reviewers' comments:

Reviewer's Responses to Questions

**Comments to the Author**

1. If the authors have adequately addressed your comments raised in a previous round of review and you feel that this manuscript is now acceptable for publication, you may indicate that here to bypass the “Comments to the Author” section, enter your conflict of interest statement in the “Confidential to Editor” section, and submit your "Accept" recommendation.

Reviewer #1: (No Response)

2. Is the manuscript technically sound, and do the data support the conclusions?

Reviewer #1: (No Response)

3. Has the statistical analysis been performed appropriately and rigorously? 

Reviewer #1: (No Response)

4. Have the authors made all data underlying the findings in their manuscript fully available?

Reviewer #1: (No Response)

5. Is the manuscript presented in an intelligible fashion and written in standard English?

Reviewer #1: (No Response)

6. Review Comments to the Author

Reviewer #1: Thank you for addressing the comments.

Minor edit required to conclusion as the children participants have been removed so need to be removed from conclusion:

"it is evident that the DETECT system has had success across three key groups of stakeholders: children, parents, and health professionals"

7. PLOS authors have the option to publish the peer review history of their article (what does this mean?). If published, this will include your full peer review and any attached files.

Reviewer #1: **Yes: **Fenella J Gill

---

## [Author Response · Author response to Decision Letter 1]

13 Jun 2022

Rebuttal to reviewer

Thank you for giving up your time to review our paper and for noting the one minor change we had missed in our previous revision.

The new text is presented in red.

Review Comments to the Author

Reviewer #1: 

Minor edit required to conclusion as the children participants have been removed so need to be removed from conclusion:

"it is evident that the DETECT system has had success across three key groups of stakeholders: children, parents, and health professionals"

Thank you for this comment, we have removed the reference to three stakeholder groups and the reference to children. The text now reads. 

(p25) … it is evident that the DETECT system has had success across two key groups of stakeholders: parents, and health professionals.

---

## [Editor Report · Decision Letter 2]

12 Aug 2022

Clinical utility and acceptability of a whole-hospital, pro-active electronic paediatric early warning system (the DETECT study): a prospective e-survey of parents and health professionals.

PONE-D-21-34519R2

Dear Dr. Carter,

We’re pleased to inform you that your manuscript has been judged scientifically suitable for publication and will be formally accepted for publication once it meets all outstanding technical requirements.

Kind regards,

Miquel Vall-llosera Camps.

Senior Editor

PLOS ONE
---

## [Editor Report · Acceptance letter]

2 Sep 2022

PONE-D-21-34519R2 

Clinical utility and acceptability of a whole-hospital, pro-active electronic paediatric early warning system (the DETECT study):  a prospective e-survey of parents and health professionals. 

Dear Dr. Carter:

I'm pleased to inform you that your manuscript has been deemed suitable for publication in PLOS ONE. Congratulations! Your manuscript is now with our production department. 

Kind regards, 

on behalf of

Dr. Miquel Vall-llosera Camps 

Staff Editor

PLOS ONE